# *DDX3Y* is likely the key spermatogenic factor in the AZFa region that contributes to human non-obstructive azoospermia

Ann-Kristin Dicke [1], Adrian Pilatz[2], Margot J. Wyrwoll[1], Margus Punab[3,4,5], Christian Ruckert[6], Liina Nagirnaja[7], Kenneth I. Aston[8], Donald F. Conrad [7], Sara Di Persio[9], Nina Neuhaus[9], Daniela Fietz[10], Maris Laan [5], Birgit Stallmeyer[1,11] & Frank Tüttelmann [1,11✉]

Non-obstructive azoospermia, the absence of sperm in the ejaculate due to disturbed spermatogenesis, represents the most severe form of male infertility. De novo microdeletions of the Y-chromosomal AZFa region are one of few well-established genetic causes for NOA and are routinely analysed in the diagnostic workup of affected men. So far, it is unclear which of the three genes located in the AZFa chromosomal region is indispensible for germ cell maturation. Here we present four different likely pathogenic loss-of-function variants in the AZFa gene *DDX3Y* identified by analysing exome sequencing data of more than 1,600 infertile men. Three of the patients underwent testicular sperm extraction and revealed the typical AZFa testicular Sertoli cell-only phenotype. One of the variants was proven to be de novo. Consequently, *DDX3Y* represents the AZFa key spermatogenic factor and screening for variants in *DDX3Y* should be included in the diagnostic workflow.

[1] Institute of Reproductive Genetics, University of Münster, 48149 Münster, Germany. [2] Clinic for Urology, Paediatric Urology and Andrology, Justus Liebig University Gießen, 35390 Gießen, Germany. [3] Andrology Centre, Tartu University Hospital, 50406 Tartu, Estonia. [4] Institute of Clinical Medicine, University of Tartu, 50406 Tartu, Estonia. [5] Institute of Biomedicine and Translational Medicine, University of Tartu, 50411 Tartu, Estonia. [6] Institute of Human Genetics, University of Münster, 48149 Münster, Germany. [7] Division of Genetics, Oregon National Primate Research Center, Oregon Health & Science University, Beaverton, OR, USA. [8] Andrology and IVF Laboratory, Department of Surgery (Urology), University of Utah School of Medicine, Salt Lake City, UT, USA. [9] Centre of Reproductive Medicine and Andrology, University Hospital Münster, 48149 Münster, Germany. [10] Institute of Veterinary Anatomy, Histology and Embryology, Justus Liebig University Gießen, 35392 Gießen, Germany. [11] These authors contributed equally: Birgit Stallmeyer, Frank Tüttelmann
✉email: frank.tuettelmann@ukmuenster.de

nfertility affects 15% of couples, and 50% of infertility cases are due to male factors[1]. About 10–20% of infertile males are diagnosed with non-obstructive azoospermia (NOA)[2–5], which represents the most severe form of male infertility and is defined as absence of sperm in the ejaculate due to impaired spermatogenesis. One of the few well-established genetic causes of NOA are microdeletions in the azoospermia factor (AZF) region on the long arm of the Y chromosome. This region contains three loci, called AZFa, AZFb, and AZFc (Fig. 1a), and the majority of these Y-chromosomal deletions originate de novo[6,7]. Of the three regions, complete deletions of the AZFa region have the most severe consequences on spermatogenesis, consistently resulting in a Sertoli cell-only (SCO) phenotype detected by testicular biopsy and histological evaluation;[8,9] in this phenotype, the seminiferous tubules show a complete absence of germ cells and only contain somatic Sertoli cells[8,9]. In contrast, men carrying complete AZFb deletions sometimes have spermatocytes – (pre)meiotic germ cells[10]. AZFc deletions are associated with a more variable phenotype ranging from severe oligo- to azoospermia[11]. Accordingly, the success rates for testicular sperm extraction (TESE) are virtually zero in men with complete AZFa or AZFb deletions and are around 50% in men carrying complete AZFc deletions[12,13].

The classical human AZFa region includes the single copy genes, USP9Y (formerly DFFRY) and DDX3Y (formerly DBY). A third gene, UTY, maps distally to the classical region and has sometimes also been described as an AZFa gene[14,15] (Fig. 1a). The relevance of each of these genes for spermatogenesis is under debate for decades[16]. USP9Y, encoding a ubiquitously expressed ubiquitin C-terminal hydrolase, has been excluded as a singular disease gene for spermatogenic failure because genomic variants, including complete gene deletions, were found to be paternally inherited[17–19]. UTY encodes a histone demethylase whose expression is not limited to testicular tissue and was only recently described to be expressed in human type A spermatogonia located at the basal membrane of seminiferous tubules[20].

Hemizygous Uty mutant male mice were reported to be fertile[21]. Finally, DDX3Y, encoding an RNA helicase with a testis-specific expression profile, appears to be the most promising candidate gene[22], as functional data support its involvement in germ cell maturation[23]. However, singular depletion of Ddx3y in mice does not impair spermatogenesis[24], and in humans, no (likely) pathogenic single nucleotide variants in DDX3Y have been described so far.

## Results

To elucidate the impact of monogenic variants in the genes located within the Y-chromosomal AZFa region, we screened whole exome sequencing (WES) data of 1,655 well-characterized men from the Male Reproductive Genomics (MERGE) study with unexplained crypto- or azoospermia for high-impact variants in USP9Y, DDX3Y, and UTY and identified three unrelated men of European ancestry carrying different LoF variants in DDX3Y. In addition, a further LoF variant in DDX3Y was identified in WES data of the Genetics of Male Infertility Initiative (GEMINI) study[25]. All four variants are absent in the gnomAD database (v2.1.1), abrogate at least the sequence of the C-terminal helicase domain (Fig. 1b), and are predicted to lead to degradation of the transcripts by nonsense-mediated decay. Interestingly, no DDX3Y LoF variants at all are listed in gnomAD, which comprises data of 67,961 men. In contrast, for USP9Y and UTY, no LoF variants were detected in the MERGE cohort, whereas in gnomAD 29 LoF variants are present in USP9Y and five LoF variants are present in UTY.

Patient M2171 belongs to a non-consanguineous German family and is a carrier of the hemizygous splice site variant c.1609+1del in DDX3Y (Fig. 1b). Segregation analysis revealed that the variant is of de novo origin, as the patient's fertile brother and a cousin on his father's side, whose relatedness to the variant carrier was confirmed by Somalier[26] (Supplementary Fig. 1), do not carry the respective variant (Fig. 2a). The variant was predicted to cause a shift of the splice donor site of intron 15 at

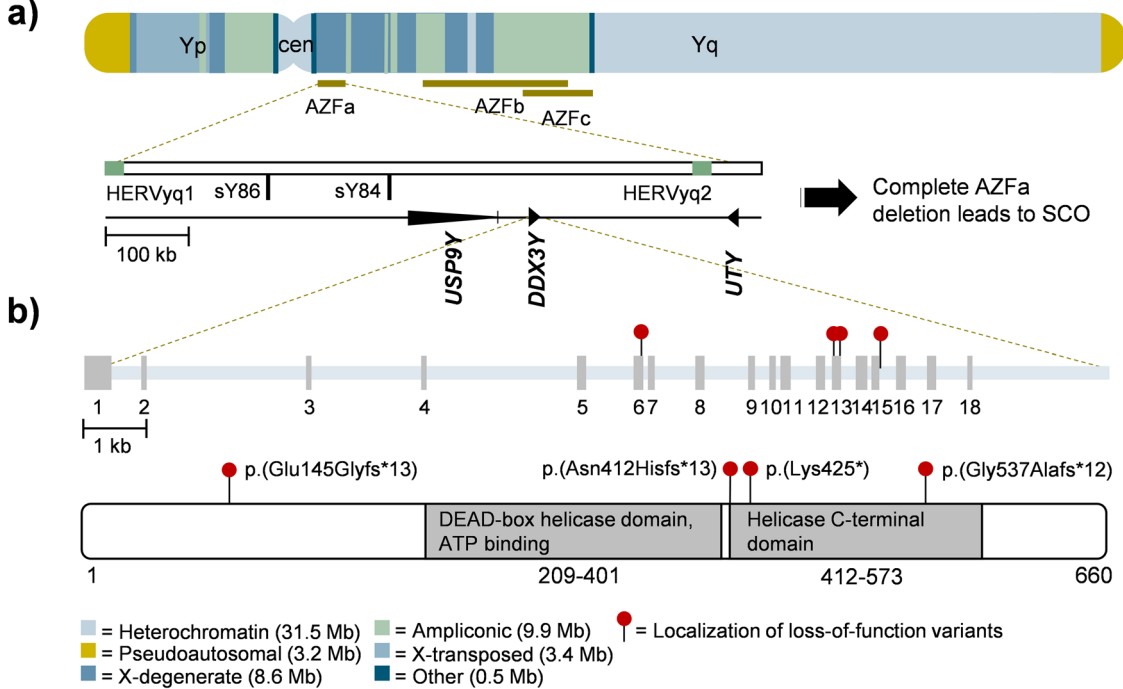

**Fig. 1 Overview of the Y-chromosomal azoospermia factor A region. a** Schematic illustration of the Y chromosome and the AZF microdeletions. The genes localizing within the AZFa region are shown as well as typical break points leading to the classic complete AZFa deletion. **b** Exon/intron structure of DDX3Y. Identified loss-of-function (LoF) variants are located in exon 6 and 13 as well as in the donor splice site of intron 15 and affect at least the sequence of the C-terminal helicase domain.

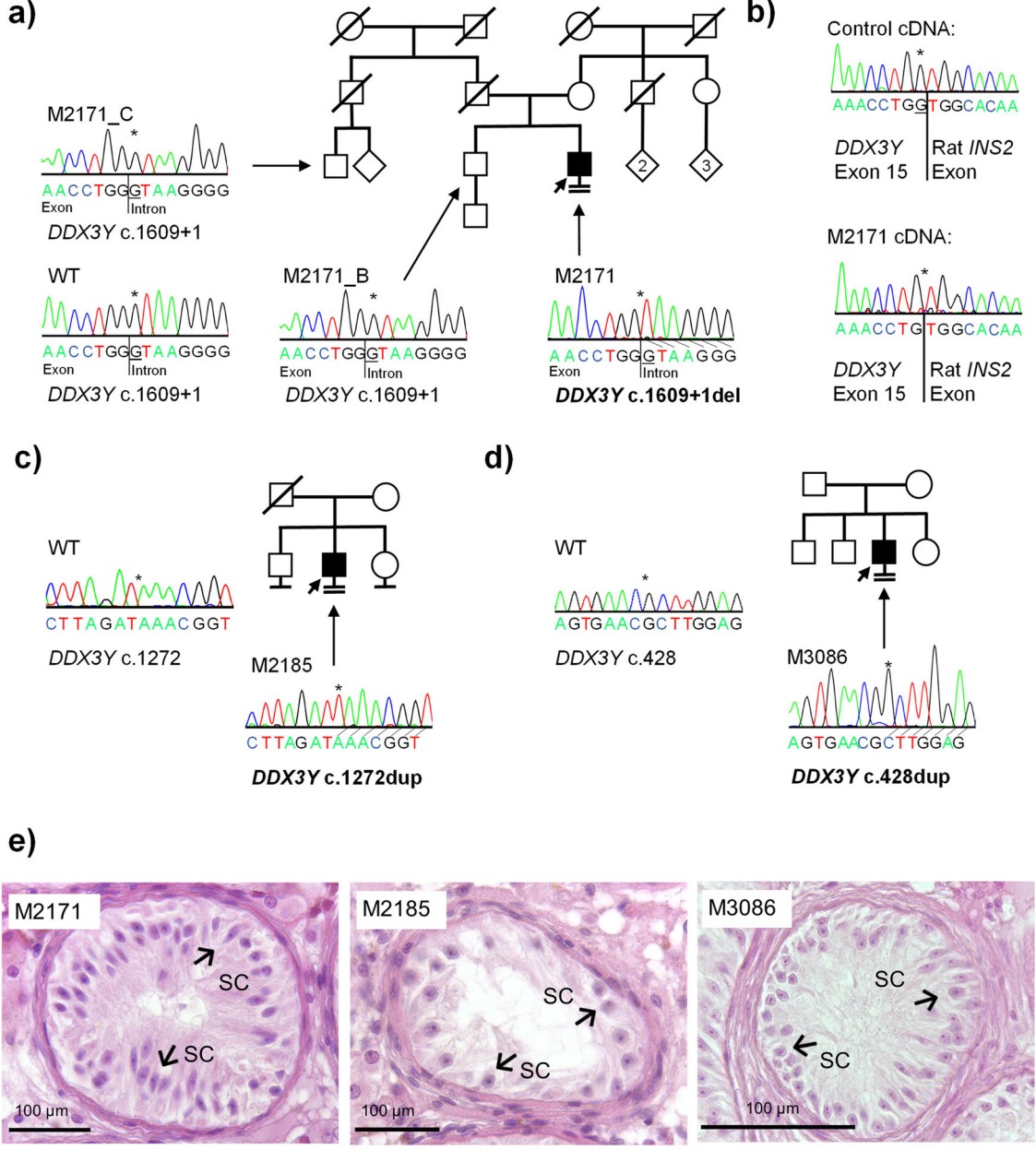

**Fig. 2 Characterization of patients M2171, M2185, and M3086. a** Pedigree of M2171 with de novo variant c.1609+1del in *DDX3Y*. **b** Sequencing of cDNA derived from splicing assay based on a minigene construct. c.1609+1del causes a shift of the splice donor site of intron 15 at position c.1609+1 to position c.1609, resulting in a frameshift. **c** Pedigree of patient M2185 with hemizygous variant c.1272dup p.(Lys425*). DNA of further family members was not available for analysis. **d** Pedigree of M3086 carrying the hemizygous variant c.428dup p.(Glu145Glyfs*13). DNA of additional family members was not available for segregation analyses. **e** Hematoxylin and eosin (HE) staining of testicular sections of M2171, M2185, and M3086, demonstrating SCO. Example Sertoli cells (SC) are indicated by arrows.

position c.1609+1 to position c.1609, resulting in a frameshift and premature stop codon (r.1609del, p.(Gly537Alafs*12)); this was confirmed in vitro by a functional splicing assay based on a minigene construct (Fig. 2b, Supplementary Fig. 2). Testicular tissue of M2171 was devoid of germ cells with only somatic Sertoli cells preserved in the seminiferous tubules (SCO) (Fig. 2e). Interestingly, patients harboring deletions of the complete AZFa region share the identical testicular phenotype. According to ACMG-AMP guidelines, the variant c.1609+1 was classified as likely pathogenic (class 4; Supplementary Table 1).

In subject M2185, who was also diagnosed with a testicular phenotype of SCO (Fig. 2e), we identified the hemizygous, likely pathogenic (class 4; Supplementary Table 1) nonsense variant c.1272dup, leading to a stop codon in exon 13 of *DDX3Y* p.(Lys425*) (Fig. 2c). No other family members with impaired fertility were reported in this non-consanguineous family, but DNA samples of male family members were not available to perform segregation analyses.

M3086, also showing a testicular phenotype of SCO (Fig. 2e), carries the hemizygous, likely pathogenic (class 4; Supplementary Table 1) frameshift variant c.428dup p.(Glu145Glyfs*13) in exon 6 of *DDX3Y* and originates from a non-consanguineous German family (Fig. 2d). No additional family members with impaired fertility were reported for this case.

Finally, in Estonian patient GEMINI-492 diagnosed with NOA, the likely pathogenic (class 4; Supplementary Table 1) frameshift

**Table 1 Clinical data of reported patients.**

| Individual | Age, nationality | Genotype | Fertility parameters | Gonadal phenotype, TESE outcome | Additional clinical features |
|---|---|---|---|---|---|
| M2171 | 31 y, German, not consanguineous | DDX3Y: g.15,028,547del c.1609+1del r.1609del p.(Gly537Alafs*12) de novo | FSH: 18.1 LH: 4.9 T: 10.58 TV: 9/12 Azoospermia | Sertoli cell-only (SCO), No sperm retrieved | Hypertonia (20 y) |
| M2185 | 37 y, German, not consanguineous | DDX3Y: g.15,027,902dup c.1272dup p.(Lys425*) | FSH: 6.1/9.8 LH: 6.7 T: 22.14 TV: 7/9 Azoospermia | SCO, No sperm retrieved | Delayed puberty, Hypertonia (29 y) |
| M3086 | 34 y, German, not consanguineous | DDX3Y: g.15,024,785dup c.428dup p.(Glu145Glyfs*13) | FSH: 30.9 LH: 5.3 T: 8.23 TV: 10/10 Azoospermia | SCO, No sperm retrieved | / |
| GEMINI-492 | 43 y, Estonian, not consanguineous | DDX3Y: g.15,027,860_15,027,861del c.1230_1231del p.(Asn412Hisfs*13) | FSH: 27.2 LH: 9.3 T: 8.4 TV: 10/10 Azoospermia | No biopsy performed | Delayed puberty |

Reference values: FSH 1–7 IU/L, LH 2–10 IU/L, T > 12 nmol/L, TV > 12 mL per testis.
TESE testicular sperm extraction, FSH follicle stimulating hormone (IU/L), LH luteinizing hormone (IU/L), T testosterone (nmol/L), TV testicular volume right/left (mL), y years.

variant c.1230_1231del p.(Asn412Hisfs*13) in exon 13 of *DDX3Y* was identified (Fig. 1b, Supplementary Fig. 3). This variant has been listed in the supplemental data of the most recent GEMINI publication[25], without presenting the clinical data. Familial data or data regarding the testicular phenotype are not available for this case.

Of note, in addition to the shared histological phenotype of SCO, all four patients with LoF variants in *DDX3Y* showed reduced testicular volume and had elevated FSH upon primary or later presentation indicative of spermatogenic failure (Table 1). Interestingly, two of the men with *DDX3Y* LoF variants had delayed puberty, and two presented with early onset hypertonia (Table 1).

According to the semi-quantitative framework of the ClinGen initiative for gene-disease validity curation, and based on the presented genetic data as well as published functional data, *DDX3Y* is assessed to have moderate clinical evidence as a disease gene for male infertility (Supplementary Data 1) and can, therefore, immediately be included in diagnostic analyses.

## Discussion

By describing four different LoF variants in *DDX3Y* in azoospermic men, this study provides sufficient evidence that *DDX3Y* is the key gene leading to spermatogenic failure observed in men with complete AZFa deletions. Three of the variant carriers underwent TESE and had a testicular phenotype of SCO. This is also the common testicular phenotype for carriers of complete AZFa deletions[8,9] and, thus, this genotype-phenotype correlation provides substantial evidence that absence or impaired function of the DDX3Y protein is the underlying cause for SCO, azoospermia, and male infertility – independent whether the mechanism is an AZFa deletion or a point variant causing loss-of-function. Further, one of the variants was verified to be of de novo origin, in line with the de novo occurrence of AZFa deletions[22] and further highlighting the recently proposed relevance of de novo genetic variants as causes of male infertility[27]. The impact of *DDX3Y* on spermatogenesis is also supported by functional in vitro data, as the impaired germ cell formation

observed in stable iPSC lines derived from patients with AZFa deletions could be rescued by the introduction of *DDX3Y*[23]. Further, the *Ddx3y* knockout is likely rescued by the *D1pas1* gene in mice, which is not translated in humans, explaining the dispensablitiy of Ddx3y in mice regarding their fertility and demonstrating the dicrepancy of mice and men concerning *DDX3Y*[24].

*DDX3Y* has a paralog on the X chromosome, *DDX3X*, and both genes encode DEAD-box containing RNA helicases that facilitate translation initiation on mRNAs[28] and have been demonstrated to be functionally exchangeable in vitro[29]. In contrast to *DDX3Y*, *DDX3X* is ubiquitously expressed and de novo occurring heterozygous variants in *DDX3X* in females are associated with a developmental disorders called *DDX3X* syndrome[30]. Hemizygous variants in men have been described in single cases. The sex bias in *DDX3X* syndrome has been explained with the inability of *DDX3Y* to compensate for the loss of *DDX3X*. Loss of DDX3X function seems to result in embryonic lethality in most male cases whereas heterozygous females are viable, but affected by neurologic symptoms[31].

Acccording to scRNAseq data of human testicular tissue, *DDX3X* as well as *DDX3Y* are expressed in spermatogonia and early spermatids[32]. However, on the protein level, DDX3Y expression is limited to spermatogonia, whereas DDX3X expression is restricted to haploid germ cells putatively explained by translational control mechanisms[33]. This suggests that the two proteins also have distinct roles in RNA metabolism of human germ cell maturation, with DDX3Y performing an important function at early stages and consequently failure of DDX3Y function causing germ cell loss and the SCO phenotype. The identification of infertile men with LoF variants in *DDX3Y*, therefore, further supports the assumption that activity of DDX3Y can also not be rescued in vivo by DDX3X, which is again a result of the celltype-specific expression profile of both proteins in testicular tissue.

Of note, a de novo single nucleotide variant had also previously been described for *USP9Y* in the context of male infertility[34]. However, the testicular tissue of this patient revealed pre-meiotic and meiotic germ cells in most seminiferous tubules with small

numbers of post-meiotic spermatids, consistent with spermatogenic arrest but not SCO. More recently, *USP9Y* was clearly excluded as an essential spermatogenesis gene because inherited deletions limited to *USP9Y* were identified in fertile men[17,18], and, accordingly, *USP9Y* is now thought to act preferentially as a fine-tuner of germ cell maturation[19].

No patients with singular deletions or single nucleotide variants have as yet been described for *UTY*, the third protein-coding gene within the AZFa region. However, the testicular phenotype of SCO is also seen in patients with classical deletions of the AZFa region encompassing only *USP9Y* and *DDX3Y* and a verified protein expression of *UTY*[22,34], which further supports *DDX3Y* as the key spermatogenic factor in the AZFa region. In addition, UTY was demonstrated to have only limited demethylase activity[35], giving rise to the question of whether the expressed protein has any critical cellular function.

In the EAA/EMQN guidelines[13], screening for AZFa deletions is recommended to start with two different markers (sY84, sY86, Fig. 1a) that map to a genomic sequence outside of the coding region for *USP9Y*, *DDX3Y*, and *UTY*; only if these markers are missing, screening is then extended to additional markers. According to our results, screening for AZF deletions should be complemented by exome sequencing to also allow detection of single exon deletions and point mutations in *DDX3Y*.

With the increasing application of WES, the number of reported monogenic causes of human male infertility has strikingly increased within the last 5 years. Many of these studies, like the first large-scale analyses in the GEMINI cohort[25], IMIGC X-chromosomal variants[36], and trio-exomes[27], aim to identify novel candidate genes and/or understand the biological pathways relevant to spermatogenesis. Still, even in large and clinically well characterized cohorts of NOA men, often only singular cases per gene are identified making it difficult to judge whether an identified gene is a valid disease gene or remains a candidate gene awaiting confirmation. This explains why the diagnostic yields reported for monogenic causes of spermatogenic failure vary between 5.4%, when only validated disease genes with at least moderate evidence based on ClinGen criteria were included[37], up to 19.3% taking all prioritized cases of a cohort into account[25]. Of note, also 6% of fertile controls were positive for a prioritized gene variant in this study, underlining the need to be careful when including genes in diagnostics to avoid false positive reports. With presenting four independent cases of infertile men with similar testicular phenotype and LoF variants in *DDX3Y*, it immediately becomes a validated disease gene for azoospermia eligible to be included in genetic diagnostics.

In summary, this study provides sufficient evidence that *DDX3Y* is the key spermatogenic factor encoded within the AZFa region and solves the conundrum of AZFa deletions that has been up for debate for more than two decades[16]. Of note, these data cannot rule out that additional non-coding sequences in the AZFa region are relevant for spermatogenesis or have an impact on gene expression and, thus, could additionally contribute to the impaired spermatogenesis observed in patients with classical AZFa deletions. In addition, independent of deciphering the genetic key component of complete AZFa deletions, *DDX3Y* is immediately a disease gene for male infertility with a clear gene-disease relationship, and patients with unexplained NOA should be screened not only for AZFa deletions but also for single nucleotide variants in *DDX3Y*. Consequently, *DDX3Y* should be included in any gene-panel for clinical diagnostic of azoospermia and should preferably be analyzed via WES because of the quickly increasing number of validated azoospermia genes[37]. Further, the clinical prognostic value for TESE success in carriers of *DDX3Y* LoF variants is very likely similar to those of men with complete AZFa deletions, i.e., virtually zero; as such, unsuccessful surgery

for sperm retrieval can be avoided in men harboring *DDX3Y* LoF variants.

## Methods

**Ethical approval.** All patients gave written informed consent for the analysis of their donated material and the evaluation of their clinical data compliant with local requirements. The MERGE study protocol was approved by the Münster Ethics Committees/Institutional Review Boards (Ref. No. Münster: 2010-578-f-S; Gießen: No. 26/11), and the approval 74/54 (last amendment 288/M-13) was released by the Research Ethics Committee of the University of Tartu, Estonia, all in accordance with the Helsinki Declaration of 1975.

**Study cohort.** The patients of the MERGE cohort were recruited in the Centre of Reproductive Medicine and Andrology (CeRA) in Münster and in the Clinic for Urology, Pediatric Urology and Andrology in Gießen. Patients with a history of radio- or chemotherapy, testicular tumors, vasectomy and hypogonadotropic hypogonadism were not included in MERGE as well as patients showing AZF deletions or chromosomal aberrations.

In addition to the MERGE cohort, we screened further whole exome sequencing (WES) data of infertile men from the International Male Infertility Genomics consortium (IMIGC, http://www.imigc.org/) for high-impact variants in *DDX3Y*. Only in the GEMINI cohort one further patient with a LoF variant was identified. This Estonian non-obstructive azoospermia (NOA) patient was recruited to the Estonian National Andrology Cohort (the ESTNAND cohort) at the Andrology Centre, Tartu University Hospital, by the managing clinician M. Punab. Recruited participants underwent a standardized andrology workup and a structured medical interview. Hormonal analysis including serum FSH, LH and testosterone were carried out and the semen quality was determined in accordance with recommendations of the World Health Organization (WHO). Cases with spermatozoa concentrations of ≤5 million/ml were screened for chromosomal aberrations and Y-chromosomal microdeletions. Known causes of NOA were excluded, e.g., cryptorchidism, testicular cancer, orchitis/epididymitis, mumps orchitis, testis trauma, karyotype abnormalities, or AZF deletions[3].

**Whole exome sequencing, variant filtering, and validation of sequence variants.** For WES of patients from the Male Reproductive Genomics (MERGE) study cohort, genomic DNA was extracted from peripheral blood leukocytes. Sample preparation and enrichment were carried out following the protocols of either Agilent's SureSelectQXT Target Enrichment kit or Twist Bioscience's Twist Human Core Exome kit. For library capturing Agilent's SureSelectXT Human All Exon Kits V4, V5 and V6 or Twist Bioscience's Human Core Exome plus RefSeq spike-in's were used. The libraries were index tagged and the quantity was assessed using the ThermoFisher Qubit while quality of the libraries was determined via Agilent's TapeStation 2200. Next generation sequencing was performed on Illumina NextSeq® 500/550 or NovaSeq® 6000 systems. Cutadapt v1.15 was used to remove remaining adapter sequences and primers following trimming[37]. Sequence reads were aligned against the reference genome GRCh37.p13 using BWA Mem v0.7.17[38]. Duplicate reads and reads that mapped to multiple locations in the genome were excluded. Single nucleotide variants and small insertions/deletions (indels) were identified and quality-filtered by GATK toolkit v3.8 with HaplotypeCaller[39]. Ensembl Variant Effect Predictor was used to annotate variants[40].

Resulting WES data of 1655 men with unexplained crypto- or azoospermia were screened for rare (minor allele frequency [MAF] ≤ 0.01 in gnomAD v2.1.1), (possibly) pathogenic variants in AZFa locus candidate genes *USP9Y* (NM_004654.4), *DDX3Y* (NM_001122665.3) and *UTY* (NM_001258249.2) and filtered based on their effect on protein sequence, including predicted loss-of-function (frameshift, stop-gain, stop/start-lost, splice site; Genome Reference Consortium Human Build 37 (GRCh37)).

WES for the ESTNAND cohort patient GEMINI-492 and prioritization of likely deleterious variants from the generated WES dataset was performed in the framework of the Genetics in Male Infertility Initiative (GEMINI) study[41] at the McDonnell Genome Institute of Washington University. Sequencing was conducted on Illumina HiSeq 4000 capturing 39.1 Mb of exome at an average coverage of 80×. Genotype calling was performed using GATK tools[39] following their best practice recommendations. The genotype callset underwent quality control procedures, including removal of positions with high missingness rates (>15%) or low coverage (<30×), high contamination (VerifyBamID freemix >5%) or low call rate (<85%). Further, genotypes with read depth (DP) < 10× and genotype quality (GQ) < 30 were excluded. Prioritization of variants was based on the population sampling probability (PSAP) software, which evaluates the probability of sampling a genotype given the pathogenicity scores and variant frequency in the general population[42].

All *DDX3Y* variants were verified by standard Sanger sequencing (for primer sequences, see Supplementary Table 2). If available, DNA from family members was sequenced for segregation analyses.

**Classification of genetic variants and gene disease curation.** All validated variants were classified according to the guidelines of the American College of

Medical Genetics and Genomics and the Association for Molecular Pathology (ACMG-AMP[43]) adapted to recent recommendations as outlined in Wyrwoll et al.[37]. We additionally assessed the clinical validity of *DDX3Y* using the semi-quantitative framework developed by the ClinGen Gene Curation working group[44].

**Minigene assay**. To functionally characterize the splice site variant c.1609+1del p.? in *DDX3Y*, an in vitro splicing assay based on a minigene construct was performed. Primers flanking exon 15 of *DDX3Y* (Supplementary Table 2, Supplementary Fig. 2b) were used to amplify the region of interest from genomic DNA of M2171 as well as a human control sample by standard PCR using 0.4 U of Phusion™ High-Fidelity DNA Polymerase. PCR products were cloned into pENTR™/D-TOPO® according to manufacturer's instructions. Subsequent gateway cloning was performed using Gateway™ LR Clonase™ Enzyme Mix and pDESTsplice as destination vector (pDESTsplice was a gift from Stefan Stamm (Addgene plasmid #32484)[45]). Human Embryonic Kidney cells, HEK293T Lenti-X (Clontech Laboratories, Inc.®) were transiently transfected (X-tremeGENE™ 9 transfection reagent) with mutant and wildtype *DDX3Y* minigenes. 24 h after transfection, total RNA was extracted using the RNeasy Plus Mini Kit (QIAGEN®) and reverse-transcribed into cDNA with the ProtoScript® II First Strand cDNA Synthesis Kit (New England Biolabs GmbH®). Amplification of the region of interest was performed using primers annealing to the rat insulin exons 3 and 4 that are part of the minigene construct (Supplementary Table 2, Supplementary Fig. 2c). RT-PCR products were separated on 2% agarose gel and sequenced.

**Hematoxylin and eosin staining of human testicular tissue**. Patients M2171 and M2185 underwent testicular biopsy at the Clinic for Urology, Pediatric Urology and Andrology in Gießen, as indicated by diagnosis of non-obstructive azoospermia (NOA) according to EAU guidelines[46] with the aim of testicular sperm extraction. After written informed consent, testicular biopsies were taken, immediately fixed in Bouin's solution, and subsequently embedded in paraffin wax. For histological analysis, 5 μm thick sections were cut, dewaxed, and stained with hematoxylin and eosin (HE) according to standard protocols. For evaluation of HE staining, a Leica DM750 microscope was used with 10-fold magnification of the ocular and 40-fold magnification of the objective lenses. Pictures were acquired with the Leica ICC50 HD camera at a primary resolution of 96 dpi. For downstream processing of images, Adobe Photoshop CS6 Extended software was used to increase the resolution to 300dpi without using interpolation procedures. Further, white balance was adjusted and PS sharpening filter was used. Finally, pictures were cropped to the desired size showing a representative detail of the testicular biopsy. Histological evaluation was performed following score count analysis[47].

**Reporting summary**. Further information on research design is available in the Nature Portfolio Reporting Summary linked to this article.

## Data availability

The biological material used in this manuscript will be available upon reasonable request. All genetic variants are submitted to ClinVar (SCV003803705 - SCV003803708). The exome data of the MERGE cohort and other IMIGC cohorts such as GEMINI are accessible by contacting the IMIGC steering committee (http://www.imigc.org/). The specific access to the exome data depends on the respective cohort and consent that varies across institutions/samples.

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

## Acknowledgements

This study relied on data from patients who gave their permission for genomic analyses, and the authors gratefully thank all participants and their family members. We furthermore thank Celeste Brennecka for language editing. This work was carried out within the frame of the German Research Foundation Clinical Research Unit 'Male Germ Cells: from Genes to Function' (DFG CRU326, grants to F.T. and N.N.). The GEMINI study has been funded by National Institutes of Health of the United States of America grants R01HD078641 (D.F.C. and K.I.A) and P50HD096723 (D.F.C.). Research of M.L. and M.P. has been supported by the Estonian Research Council grant PRG1021.

## Author contributions

Study conceptualization: B.S. and F.T. Data curation: A.-K.D. and B.S. Funding acquisition: F.T. Investigation: A.-K.D., A.P., M.J.W., M.P., C.R., L.N., K.I.A., D.F.C., S.D.P., N.N., D.F., M.L., B.S., F.T. Visualization: A.-K.D. Writing of original draft: A.-K.D. and B.S. Review and editing: A.-K.D., A.P., M.J.W., M.P., C.R., L.N., K.I.A., D.F.C., S.D.P., N.N., D.F., M.L., B.S., F.T. All authors revised and approved the final version of the manuscript.

## Funding

## Competing interests

The authors declare no competing interests.
