## [Peer Review File · Communications Biology]

Reviewers' comments:

Reviewer #1 (Remarks to the Author):

Dicke and colleagues aimed to identify genetic mutations within the AZFa region (which is located in the Y chromosome and whose deletion is associated with non-obstructive human azoospermia) in a large cohort of male infertility patients due to spermatogenic failure. They generated and analysed whole exome sequencing (WES) data from their study cohort and described loss-of-function variants in the coding region of the DDX3Y gene that explained the infertility of four patients, confirming that one of them was a de novo mutation. The authors concluded that such gene is most likely the key spermatogenic factor in the AZFa region, proposing that this locus should be screened for point mutations in the clinical evaluation of azoospermic men seeking to father a biological child through assisted reproductive techniques.

I must say that the study design is appropriate, and the results consistent and brilliantly organised and discussed, which is not strange considering the high expertise of this research group in the field of reproductive genetics. Therefore, I have only a few minor comments to make:

1) Did the authors check if both the "Male Infertility Genomic Consortium (MIGC) database" and the "Infertility Disease Database (IDDB)" have entries including DDX3Y mutations? It would be interesting to know whether DDX3Y variants related to male infertility have been previously annotated in such databases.

2) Did the carriers of DDX3Y mutations have a similar morphological affectation of the testis biopsies? If so, do the authors consider that the identified variants have the same pathogenic effect?

3) There is a paralog of DDX3Y in the X-chromosome (i.e. DDX3X), which shares most of the coding sequence and is reported to have a similar function. I consider that this information should be included and discussed in the manuscript (e.g. why is the loss of function of DDX3Y not compensated by DDX3X?).

4) The reported single-cell expression of DDX3Y in the testis suggests a more relevant role of this gene in late spermatids than in earlier stages. How would the authors explain this fact?

Reviewer #2 (Remarks to the Author):

Dicke et al. suggest that they have solved the conundrum of AZFa genetics by finding that DDX3Y is the key spermatogenic factor.

Indeed they present compelling evidence for this suggestion:

1. Based on WES of 1,655 men with unexplained crypto- or azoospermia screened for rare (minor allele frequency [MAF] ≤ 0.01 in gnomAD v2.1.1), they identified different loss of function pathogenic variants in DDX3Y in four patients. Three of the patients had testicular tissue devoid of germ cells, with only somatic Sertoli cells preserved in the seminiferous tubules (SCO), similar to the testicular phenotype of patients harboring deletions of the complete AZFa region. One patient was diagnosed with NOA. SCO is also typical for carriers of complete AZFa.

2. The other two genes in the AZFa deletions, USP9Y and UTY, do not contribute to male infertility because inherited deletions limited to USP9Y were identified in fertile men (references 6,19). SCO is also seen in patients with partial deletions of the AZFa region encompassing only USP9Y and DDX3Y and a verified protein expression of UTY (references 4,27). Additionally, a weaker suggestion to the involvement of DDX3Y and not USP9Y and UTY in male infertility is the finding that no DDX3Y loss of function (LoF) variants are listed in gnomAD (a database of 67,961 men). In contrast, USP9Y and UTY present 29 and 5 LoF variants, respectively.

3. They discuss the function of DDX3Y on spermatogenesis: in vitro the impaired germ cell formation

observed in stable iPSC lines derived from patients with AZFa deletions could be rescued by the introduction of DDX3Y (reference 22).

However, more caution should be taken for such a definite suggestion. This is because mice deleted for Ddx3y are fertile. Dicke et al. suggest that the dispensability of Ddx3y knockout is likely explained by the rescue of the D1pas1 gene in mice, which is not translated in humans. However, the paper describing the Ddx3y knockout mice (reference 23) states: "the spermatogenic failure in D1pas1 KO mice is milder than that in AZFa-deleted SCOS patients. These reports suggest that mouse D1pas1 alone is not sufficient to function as a substitute for human AZFa genes."

Thus, it is possible that additional sequences, not necessarily in coding sequences, in the AZFa region contribute to sperm development. For this reason, the EAA/EMQN guidelines for screening for AZFa deletions should not be revised. However, the study does provide sufficient evidence that DDX3Y is the key gene leading to spermatogenic failure and should be included in diagnostic analyses, preferentially by complete sequencing.

Point-by-point response to the referees' comments

Title: Having solved the conundrum of AZFa genetics – DDX3Y is the key spermatogenic factor

Manuscript number: COMMSBIO-22-3964-T

Revision Version: 1

Editor's Decision Received Date: 05.01.2023

Dear reviewers,

Thank you very much for reviewing our manuscript. We are sincerely grateful for the overall positive comments on our study and have critically revised the manuscript taking into account all points raised. We also followed the formal instructions and formatted our manuscript accordingly. Please find below the point by point answers to your comments.

Reviewer 1:

Dicke and colleagues aimed to identify genetic mutations within the AZFa region (which is located in the Y chromosome and whose deletion is associated with non-obstructive human azoospermia) in a large cohort of male infertility patients due to spermatogenic failure. They generated and analysed whole exome sequencing (WES) data from their study cohort and described loss-of-function variants in the coding region of the DDX3Y gene that explained the infertility of four patients, confirming that one of them was a *de novo* mutation. The authors concluded that such gene is most likely the key spermatogenic factor in the AZFa region, proposing that this locus should be screened for point mutations in the clinical evaluation of azoospermic men seeking to father a biological child through assisted reproductive techniques.

I must say that the study design is appropriate, and the results consistent and brilliantly organised and discussed, which is not strange considering the high expertise of this research group in the field of reproductive genetics. Therefore, I have only a few minor comments to make:

Thank you very much for your positive comments on our manuscript.

1) Did the authors check if both the "Male Infertility Genomic Consortium (IMIGC) database" and the "Infertility Disease Database (IDDB)" have entries including DDX3Y mutations? It would be interesting to know whether *DDX3Y* variants related to male infertility have been previously annotated in such databases.

We are closely working together with all members of IMIGC and screened the WES data of IMIGC for additional variants in *DDX3Y*. Accordingly, the Estonian case GEMINI-492 is from one of the IMIGC cohorts and was included in the manuscript. In addition, this variant is also listed in the supplemental data the most recent publication including IMIGC data (Nagirnaja L. *et al.* Diverse Monogenic Subforms of Human Spermatogenic Failure. PMID: 36572685), however without presenting any clinical data. At the time of submission, this manuscript was not yet accepted. We now describe the interaction with IMIGC on l. 223ff.

We furthermore checked the Infertility Disease Database, however no entries regarding variants in *DDX3Y* were listed.

2) Did the carriers of DDX3Y mutations have a similar morphological affectation of the testis biopsies? If so, do the authors consider that the identified variants have the same pathogenic effect?

Three of the four presented patients with LoF variants in *DDX3Y* underwent TESE and consistently show a Sertoli cell-only phenotype in HE staining of testicular sections. As all variants are predicted to cause a complete absence of *DDX3Y* and, thus, a loss of protein function in the testis, we conclude that they share the same underlying pathogenic effect. Since the testicular phenotype of SCO is also consistently observed in patients with complete AZFa deletion, this phenotypic overlap is substantial evidence for *DDX3Y* being the key spermatogenic factor within the AZFa region. The testicular phenotype of SCO is described for each of the patients in the results and depicted in Figure 2E. Because this might not have been entirely clear, we explicitly state this now on l. 116 and also l. 128 ff.

3) There is a paralog of *DDX3Y* in the X-chromosome (i.e. *DDX3X*), which shares most of the coding sequence and is reported to have a similar function. I consider that this information should be included and discussed in the manuscript (e.g. why is the loss of function of *DDX3Y* not compensated by *DDX3X*?).

Thank you for this valuable suggestion. We absolutely agree that it is important to discuss the role and function of *DDX3X* in our context and, therefore, included an additional section in the discussion (l. 141ff).

4) The reported single-cell expression of *DDX3Y* in the testis suggests a more relevant role of this gene in late spermatids than in earlier stages. How would the authors explain this fact?

Thank you very much for raising this interesting point. *DDX3Y* transcripts are indeed expressed in spermatogonia and haploid germ cells. However, it has been demonstrated that on the protein level, *DDX3Y* expression is restricted to spermatogonia and in contrast *DDX3X* expression is limited to haploid germ cells. This observation has been explained with translational control mechanism (PMID: 15294876). We included the information on the transcriptional and translational expression pattern of *DDX3Y* and *DDX3X* in the discussion (l. 150ff).

Reviewer 2:

Dicke et al. suggest that they have solved the conundrum of AZFa genetics by finding that *DDX3Y* is the key spermatogenic factor.

Indeed they present compelling evidence for this suggestion:

Thank you very much for this positive statement on our manuscript. Please find our answers on your comments below:

1) However, more caution should be taken for such a definite suggestion. This is because mice deleted for *Ddx3y* are fertile. Dicke et al. suggest that the dispensability of *Ddx3y* knockout is likely explained by the rescue of the *D1pas1* gene in mice, which is not translated in humans. However, the paper describing the *Ddx3y* knockout mice (reference 23) states: “the spermatogenic failure in *D1pas1* KO mice is milder than that in AZFa-deleted SCOS patients. These reports suggest that mouse *D1pas1* alone is not sufficient to function as a substitute for

human AZFa genes.” Thus, it is possible that additional sequences, not necessarily in coding sequences, in the AZFa region contribute to sperm development.

Thank you for this valuable comment. In the mentioned manuscript the authors also state that “it will be useful to conduct studies to determine whether DDX3Y and D1pas1 compensate for each other in mice by generating DDX3Y and D1pas1 double KO mice” and this would be a valuable approach to determine whether this double KO mice share the testicular phenotype of patients with AZFa deletions. We also agree that it is important to be careful with definite statements. Therefore, we describe *DDX3Y* using the term “key spermatogenic factor” or “genetic key component” to underline that *DDX3Y* is the most important factor within this region causing the observed SCO phenotype, but at the same time we tried to avoid phrases of exclusivity or to describe *DDX3Y* as “the AZFa gene”.

We furthermore added an additional short section in the discussion to emphasise that we cannot exclude a possible pathogenic role of other sequence variations (l. 196ff).

2) For this reason, the EAA/EMQN guidelines for screening for AZFa deletions should not be revised. However, the study does provide sufficient evidence that *DDX3Y* is the key gene leading to spermatogenic failure and should be included in diagnostic analyses, preferentially by complete sequencing.

While we indeed expect that guidelines will change in the future to move toward sequencing based approaches, we agree that it might be too early for such a statement. Thus, we took up your suggestion and rephrased the section (l. 176-178).

Editor’s comments:

We therefore invite you to revise and resubmit your manuscript, taking into account the points raised. In particular, do please include some additional discussion regarding the context of your findings within the broader framework of Reference #25 (which has now been published: <https://www.nature.com/articles/s41467-022-35661-z>).

Thank you very much for inviting us to resubmit a revised version of the manuscript. We gladly take up your proposal to discuss our findings in a broader context and, to this end, added a new paragraph (l. 179ff).

Further changes:

We better reflect the ongoing discussion on whether *UTY* should be considered an AZFa gene by slightly revising l. 63-64.

We added more detail to the methods concerning how digital histological images were taken and processed (l. 279ff).